# 1  TiP-Leaf: A dataset of leaf traits across vegetation types
# 2  on the Tibetan Plateau

Yili Jin[1], Haoyan Wang[1], Jie Xia[1], Jian Ni[1], Kai Li[1], Ying Hou[1], Jing Hu[1], Linfeng Wei[1],
Kai Wu[1], Haojun Xia[1], Borui Zhou[1]
[1]College of Chemistry and Life Sciences, Zhejiang Normal University, Jinhua 321004, China
**Correspondence:** Jian Ni (nijian@zjnu.edu.cn)
**Abstract.** Functional trait databases are emerging as a crucial tool for a wide range of ecological studies
including the next-generation vegetation modeling across the world. However, few large-scale studies
have been reported on plant traits in the Tibetan Plateau (TP), the cradle of East Asian flora and fauna
with specific alpine ecosystems, no report on plant trait databases could be found. Here an extensive
dataset of 11 leaf functional traits (TiP-Leaf) for mainly herbs and shrubs and a few trees on the TP was
compiled through field surveys. The TiP-Leaf dataset, compiled from 336 sites distributed mainly in the
plateau surface and the northern margin of the TP across alpine and temperate vegetation regions and
sampled from 2018 to 2021, contains 1692 morphological trait measurements of leaf thickness, leaf fresh
weight, leaf dry weight, leaf dry-matter content, leaf water content, leaf area, specific leaf area and leaf
mass per area and 1645 chemical element trait measurements of leaf carbon, nitrogen and phosphorus
contents. Thus, 468 species belonging to 184 genera and 51 families were obtained and measured. In
addition to leaf trait measurements, geographic coordinates, bioclimate variables, disturbance intensity
and vegetation types of each site were also recorded. The dataset could provide solid data support for
effectively quantifying the modern ecological features of alpine ecosystems, further evaluating the
response of alpine ecosystem to climate change and human disturbances and improving the next-
generation vegetation model. It could be a great contribution to the regional and global plant trait
databases. The dataset is available from the National Tibetan Plateau Data Center (TPDC; Jin et al., 2022;
https://doi.org/10.11888/Terre.tpdc.272516).

## 25  1 Introduction

Plant traits of morphological, anatomical, physiological and phenological characteristics respond to
changes in the living environment, affect ecosystem functions (Díaz & Cabido, 2001) and drive species



coexistence under environmental constraints (Violle et al., 2007). Over the past three decades, a growing
body of trait analyses has quantified the global and regional distribution patterns of key functional traits,
such as leaf (Reich & Oleksyn, 2004; Wright et al., 2004), seed size (Moles et al., 2007), plant height
(Moles et al., 2009), wood (Chave et al., 2009), plant form and function (Diaz et al., 2016), root (Ma et
al., 2018) and flower (Roddy et al., 2021). Such studies have successfully linked plant traits with
environmental change (Meng et al., 2009, 2015; Myers-Smith et al., 2019; Maes et al., 2020; Wang et
al., 2022), natural and anthropogenic disturbances (Diaz et al., 2007) and ecosystem functions
(Reichstein et al., 2014). Findings from plant trait–environment–ecosystem function interaction could be
further utilised to map the spatial pattern of plant traits (Butler et al., 2018), build the next-generation of
vegetation model (Berzaghi et al., 2020), predict vegetation distribution (van Bodegom et al., 2014) and
function (Wang et al., 2017) and be incorporated into Earth system model (Wullschleger et al., 2014).
New insights into ecosystem traits (He et al., 2019) and trait network (He et al., 2020) are bridging
multiple dimensions of biology, macroecology and geoscience. All these works require global and
regional plant trait databases, such as the TRY (Kattge et al., 2011, 2020), Growth-Form (Taseski et al.,
2019), Global Inventory of Floras and Traits (Weigelt et al., 2019), Fine-Root (Iversen et al., 2017),
GRoot (Guerrero-Ramírez et al., 2021), and tundra traits (Bjorkman et al., 2018), Plant Trait for
Mediterranean Basin Species (BROT) (Tavşanoğlu & Pausas, 2018), China traits (Wang et al., 2018),
Aus-Traits (Falster et al., 2021) and LT-Brazil (Mariano et al., 2021).
Plant trait databases across various biomes at global, continental and regional scales have been
largely raised, even in some remote areas with logistical difficulties, including the tundra (Bjorkman et
al., 2018) and tropical regions (Mariano et al., 2021). However, the Earth still has under-sampled regions.
The Tibetan Plateau (TP), known as the world's "Third Pole" and "Asia Water Tower" and the cradle of
the East Asian flora, is the most under-representative region in global and regional plant trait databases.
The first version of Chinese plant trait database (Wang et al., 2018) has no data from the TP and the
global plant trait database TRY (Kattge et al., 2020) has few collections from various sources with non-
systematic sampling. Field-based, small regional studies of plant functional traits on the TP were also
limited (Luo et al., 2005; He et al., 2006; He et al., 2010; Geng et al., 2014; Wang et al., 2020; Xu et al.,
2021), where the sampling sites have been mostly along the main roads in East TP. Plant trait records
from Central to West TP are very rare. However, the TP has the richest temperate alpine flora (Ding et



al., 2020) and most abundant plant diversity in the world (Wang & Hong, 2022). It was also an
evolutionary cradle for herbaceous genera of China (Lu et al., 2018). The uplift of the TP and its unique
alpine vegetation are important to the monsoon climate system and vegetation of East Asia (Chang et al.,
1983) and regional and global climate change studies (Piao et al., 2019).

As the largest and highest plateau in the world, the TP has not only changed the regional and global

climate system, geological structure, topography and hydrology (Yao et al., 2012) but also strongly
influenced the evolution of the flora, fauna and biodiversity (Ding et al., 2020). It has 8876 vascular
species from 1371 genera and 211 families, including 6475 herbaceous and 2401 woody plants, of which
1706 were endemic to the TP (Yan et al., 2013). Here has three biodiversity hotspots of the world (Sloan
et al., 2014; Wang & Hong, 2022). Vegetation changes from the southeast to northwest, from lowland
broadleaved evergreen forests including tropical rainforest and subtropical evergreen forest, montane
mixed evergreen and deciduous forest, subalpine conifer forest to alpine shrubland, meadow, steppe and
desert, along an annual precipitation gradient from ca. 3000 mm to 50 mm (Chang, 1983). The unique
alpine vegetation looks like the arctic tundra in physiognomy but has different species composition. The
plateau has amplified changes in climates (Chen et al., 2015) and rapid climate change has led to
profound changes in alpine species and ecosystems (Zhang et al., 2015; Piao et al., 2019). Plant traits, as
the link amongst species, environment and ecosystem functions, are a best tool to study the impacts of
climate change on vegetation. Therefore, the establishment of TP plant trait database and further analysis
of plant trait–environment–ecosystem function relationships are of great significance to understanding
the future change and sustainable development of the unique alpine vegetation on the roof of the world.

Here, a TP leaf trait dataset (TiP-Leaf) was established and 11 leaf traits from 468 species of 1692

leaf samples were collected from 336 sites across five of six vegetation types on the TP. Climate data of
the sites were also provided. This dataset is not only an update of the Chinese plant trait database but
also a great contribution to the global trait database.
**2 Study areas**

The leaf traits of dominant and common plant species of the TP distributed mostly across the plateau

surface were sampled and measured from July to August in the summer of 2018–2021. Vegetation
surveys were conducted in four regions (Fig. 1), the source area of the Three Rivers in Qinghai Province



in Northeast TP (2018); Southern Xizang Autonomous Region in Southeast and Middle TP (2019); Ngari
Prefecture in Northwest TP and the Qaidam Basin of Northeast TP (2020); and the Qilian Mountains, the
Altun Mountains and the Kunlun Mountains in the northern margin of the TP, passing through the
southern margin of the Tarim Basin in Xinjiang Autonomous Region (2021).

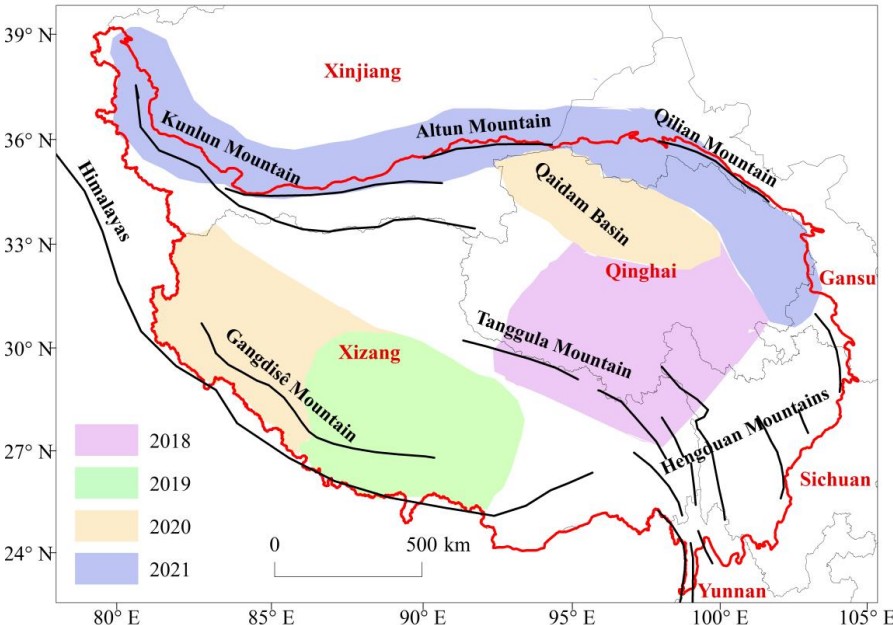


**Figure 1.** Location and administrative division of the TP. The red line indicates the boundary of the TP in China,
which involves six administrative divisions (light black lines): Xizang, Qinghai, Sichuan, Xinjiang, Gansu and
Yunnan. The bold black lines represent important mountains. Four blocks with different colours represent the
approximate areas of four investigations conducted in various years. The background map is from the Chinese
National Bureau of Surveying and Mapping.
**3 Materials and methods**
**3.1 Sampling sites**

Taking the zonal vegetation types and the precipitation gradient into account, 336 sites (Fig. 2) were

selected to investigate the vegetation with less grazing and other anthropogenic disturbances. Shrubby
and herbaceous vegetation were mainly selected (332 sites), along with forest vegetation in four sites
(but removed for further analysis). In each site, 1–3 plots were set up to survey the species composition,
abundance, coverage and plant height. The plot areas for herbaceous vegetation, shrubby vegetation and
forest vegetation were 1 m × 1 m, 2 m × 2 m or 5 m × 5 m and 10 m × 10 m, respectively. Geographical
locations, natural and human disturbances, and vegetation types were also recorded (Jin et al., 2022).
Plant leaf samples were picked up and the leaf traits were measured. Root samples were obtained using
soil pit method. The root traits were also measured but not shown in this paper.

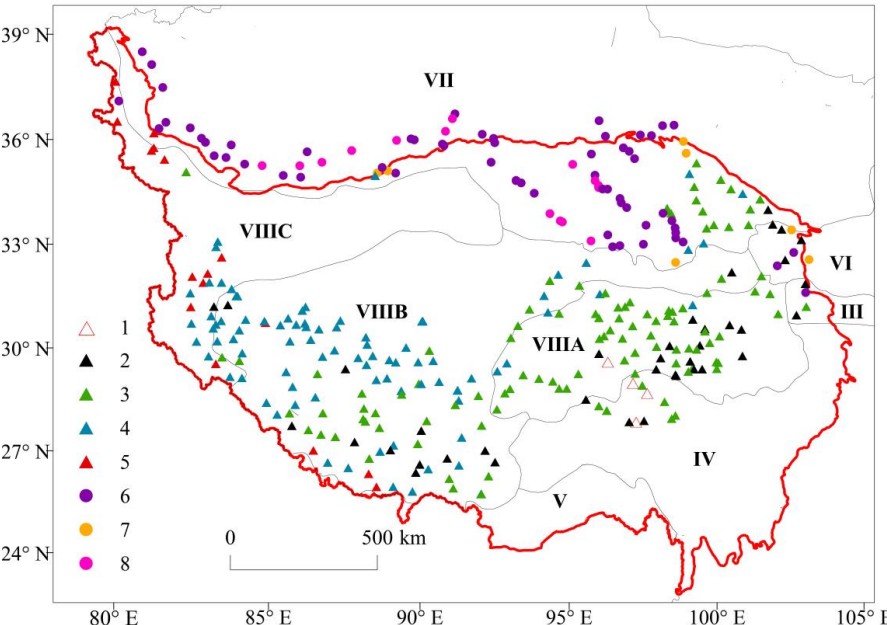


**Figure 2.** Site distribution of the TiP-Leaf dataset. Vegetation regions were extracted from the vegetation
regionalisation of China (ECVMC, 2007b). III, Warm Temperate Deciduous Broadleaf Forest Region; IV,
Subtropical Broadleaf Evergreen Forest Region; V, Tropical Monsoon Rain Forest and Rain Forest Region; VI,
Temperate Steppe Region; VII, Temperate Desert Region; VIII, TP Alpine Vegetation Region. VIIIA, East TP Alpine
Scrub and Alpine Meadow Subregion; VIIIB, Middle TP Alpine Steppe Subregion; VIIIC, Northwest TP Alpine
Desert Subregion. Vegetation types were classified on the basis of field records. Numbers indicate the vegetation
types recorded in the field. 1, coniferous forest; 2, alpine shrubland; 3, alpine meadow; 4, alpine steppe; 5, alpine
desert; 6, temperate desert; 7, temperate steppe; 8, temperate meadow. The background map is from the Chinese
National Bureau of Surveying and Mapping.

The vegetation type of the TP was classified on the basis of field records into eight types: high-cold

(alpine) shrubland, meadow, steppe and desert, temperate steppe, meadow and desert, and coniferous
forest (Fig. 2). Alpine shrubland is dominated by evergreen broad-leaved shrubs (*Rhododendron*),
deciduous broad-leaved shrubs (*Salix*, *Dasiphora*, *Sibiraea*) and evergreen coniferous shrubs (*Juniperus*),
distributed in the cold and semi-humid Southeast TP (ca. 600-1000 mm/year). Alpine meadow is widely



developed in East TP, where cold and wet climates dominate (ca. 600 mm/year), dominated by several
*Kobresia* species and mixed in with perennial forbs and cushion plants. Alpine steppe is in the middle
TP, with a large continuous distribution, adapted to the cold and semi-dry continental climate (ca. 200
mm/year) and mainly composed of *Stipa* and *Artemisia*. Alpine desert is mainly distributed in Northwest
TP, where the climate is extremely continental (ca. 50 mm/year), dominated by *Krascheninnikovia*
*compacta* and *Ajania tibetica*. Temperate meadow, steppe and desert are distributed in the northern
margin of the TP and Qaidam Basin of Northeast TP, where elevations are lower and the climate is
relatively dry, dominated by several xerophytes, especially *Haloxylon ammodendron*, *Halogeton*
*glomeratus*, *Phragmites australis*, *Ephedra*, *Kalidium*, *Calligonum* and *Tamarix*. Subalpine coniferous
(*Abies* and *Picea*) forests are found on the southeast and east margins. Therefore, the vegetation of the
plateau is distributed along a transitional gradient from southeast to northwest, arraying from subalpine
forests, alpine meadow and scrub, through alpine steppe and temperate desert to alpine desert. The alpine
vegetation was usually called high-cold vegetation in the vegetation classification of China [Editorial
Committee of Vegetation Map of China (ECVMC), The Chinese Academy of Sciences, 2007a]. Lowland
tropical and montane subtropical evergreen forests do not exist in the sampling area, hence not included
in this study.
Each site was also assigned to a vegetation region on the basis of the vegetation regionalisation of
China (ECVMC, 2007b). The TP has six vegetation regions: Alpine Vegetation Region, Temperate
Steppe Region, Temperate Desert Region, Warm Temperate Deciduous Broadleaf Forest Region,
Subtropical Broadleaf Evergreen Forest Region, and Tropical Monsoon Rain Forest and Rain Forest
Region. The sampling sites were mainly concentrated in the Alpine Vegetation Region. Therefore, in
accordance with the degree of drought, TP vegetation was further divided into three subregions from
southeast to northwest: East TP Alpine Scrub and Alpine Meadow Subregion, Middle TP Alpine Steppe
Subregion and Northwest TP Alpine Desert Subregion.
The plant name was determined in accordance with *Flora of China* (Editorial Committee of Flora
of China, 1959-2004), *Flora of Qinghai* (Editorial Committee of Flora of Qinghai, 1996-1999), *Flora of*
*Xizang* (Comprehensive Scientific Investigation Team of Qinghai-Tibet Plateau, Chinese Academy of
Sciences, 1985-1987), *Flora of Gansu* (Editorial Committee of flora of Gansu, 2005), *Flora of Xinjiang*
(Editorial Committee of flora of Xinjiang, 1993-1996) and *Flora of Deserts* in China (Liu, 1985). The

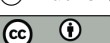

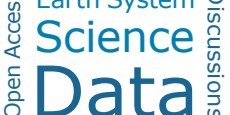

final species correction was based on the iPlant website (http://www.iplant.cn/), which merged all of the
information from the Chinese and English versions of *Flora of China* on the basis of APG IV
classification (Angiosperm Phylogeny Group, 2016).
**3.2 Leaf trait measurements**
At each site, 2–3 mature and disease-free complete leaves from each individual of dominant and
common plant species were collected, and at least 30 individuals were selected to meet the needs of trait
measurement and element analysis. When the single leaf was small, micro or leptophyllous, 100–200
leaves were picked. In total, 11 leaf functional traits (leaf thickness, LT; fresh weight, FW; dry weight,
DW; leaf dry-matter content, LDMC; leaf water content, LWC; leaf area, LA; specific leaf area, SLA;
leaf mass per area, LMA; leaf carbon content, LCC; leaf nitrogen content, LNC; and leaf phosphorus
content, LPC) were measured and calculated on the basis of the handbook of standardised measurement
for plant functional traits worldwide (Cornelissen et al., 2003; Pérez-Harguindeguy et al., 2013).
LT (mm) was measured on sampling day by using Vernier callipers with an accuracy of 0.01 mm.
The thickness in the middle of the vein and margin of each leaf was measured and then the average of
the five leaves was taken as the LT of a species. In addition to LT, 20–30 leaves for normal-leaved plants
and 100–200 leaves for small- to leptophyll-leaved plants were generally selected for other trait variable
measurements. FW (g) was obtained by weighing with 1/100 electronic balance. The fresh leaves were
then oven dried at 75 °C for 48–72 hours to obtain the DW (g). LDMC was measured as follows: LDMC
($g·g^{-1}$) = DW/FW. LA was measured using a scanner (EPSON Perfection V 700 Photo Scanner) and a
software (WinFOLIA Pro, Canada). SLA and LMA were measured as follows: SLA ($cm^2·g^{-1}$) = LA/DW
and LMA ($g·m^{-2}$) = DW/LA. The dried leaves were further used for chemical analysis. LCC ($mg·g^{-1}$),
LNC ($mg·g^{-1}$) and LPC ($mg·g^{-1}$) were determined by outside-temperature hot potassium dichromate
oxidation–volumetric method, distillation–titration method and vanadium molybdate yellow colorimetric
method, respectively.
**3.3 Climate data**
Climate data of each sampling site were extracted from the climate and bioclimate datasets of China
(Cheng et al., submitted; Wei et al., 2022). China's climate dataset consists of three variables (monthly
temperature, precipitation and sunshine percentage) that were averaged from long-term records from



1981 to 2010 at 2152 meteorological stations across China (China Meteorological Data Service Centre,
http://data.cma.cn). These three climate factors and the absolute maximum and minimum temperatures
during the 30 years of 1981–2010 were interpolated into 1 km grid cells by using a surface fitting
technique of thin-plate smoothing spline (ANUSPLIN version 4.4, Hutchinson & Xu, 2013; Xu &
Hutchinson, 2013) that took the impact of elevation on climates into account on the basis of the digital
elevation model of the Shuttle Radar Topography Mission (SRTM) (Farr et al., 2007). The interpolated
climate data were used to drive a bioclimate software (Gallego-Sala et al., 2010) to calculate the mean
annual temperature (MAT), mean temperature of the coldest month (MTCO), mean temperature of the
warmest month (MTWA), annual growing degree days above 0 °C ($GDD_0$) and 5 °C ($GDD_5$), mean
annual precipitation (MAP), growing season precipitation (GP), annual drought index (1-AET/PET) and
annual moisture index (MAP/PET), where AET is annual actual evapotranspiration and PET is annual
potential evapotranspiration.
**3.4 Data analysis**
Six key leaf functional traits (LT, LDMC, SLA, LCC, LNC and LPC) were selected in this paper
for further simple statistical analyses. The mean, minimum, maximum, standard deviation (SD) and
coefficient variation of traits at each site were calculated. The linear relationships between leaf traits of
site average were analysed.
**4 Data description of sampling sites**
**4.1 Spatial distribution of sites**
Eleven key plant leaf traits of 1692 individuals of 468 species from 336 sites were measured (Fig.
1 and 2).
The sampling sites were located in the northeast, middle to southwest, and north margin of the TP,
along with 145 sites in Xizang, 121 sites in Qinghai, 43 sites in Xinjiang, 16 sites in Gansu and 11 sites
in Sichuan (Fig. 1). Southeast TP, where forest ecosystems are distributed, has few plant trait data.
However, field measurements are being conducted in the Hengduan Mountains to measure the leaf, twig
and root traits of dominant and common trees and shrubs. Other ecologists have worked on some parts
of this region to perform leaf and other trait studies (Luo et al., 2005; Shi et al., 2012; Vandvik et al.,





2020; Xu et al., 2021). The Hoh Xil dead zone in Central North to Northwest TP is logistically not
accessible during the plant growing season when the frozen ground is melting. Therefore, the plant trait
data have been not available up to now.

**4.2 Altitudinal range of sites**

The altitudinal range of the sampling sites was between 805–5343 m, in which 69.3% of the sites

were located in the high altitudes (> 3500 m), 18.5% of the sites were located in the Qaidam Basin and
the East Qinghai with lower altitudes (2500–3500 m) and 12.2% of the sites were located in the northern
margin of the TP with lowest altitudes (< 2500 m).

**4.3 Vegetation types of sites**

In accordance with the field records, the vegetation was divided into eight types, along with 108

sites in alpine meadow, 87 sites in alpine steppe, 61 sites in temperate desert, 38 sites in alpine shrubland,
16 sites in alpine desert, 15 sites in temperate meadow, seven sites in temperate steppe and four sites in
forest. In addition, the number of sites in the TP Alpine Vegetation Region was the most abundant (63.1%)
and its three subregions, namely the the Middle TP Alpine Steppe Subregion (33.9%), the East TP Alpine
Scrub and Alpine Meadow Subregion (20.8%) and the Northwest TP Alpine Desert Subregion (8.3%),
followed by the Temperate Desert Region (29.4%) and other vegetation regions (7.5%), which are
Subtropical Evergreen Broadleaved Forest Region (3.9%), Temperate Steppe Region (2.4%) and Warm
Temperate Deciduous Broadleaved Forest Region (1.2%), as shown in Fig. 2.

**5 Data description of species and traits**

**5.1 plant species**

A total of 1692 leaf samples were collected and measured in the TiP-Leaf dataset, including 468

species belonging to 184 genera in 51 families (amongst them, 17 samples were identified as genera, six
samples were identified as families and one sample could not be identified). Some species were
frequently sampled. For example, *Kobresia pygmaea* occurred 52 times, mainly in East and South TP;
*Stipa purpurea* occurred 47 times, mainly in Middle and West TP; and *Potentilla bifurca* occurred 41
times, mainly in South, West and Northeast TP. However, in some sites, only one or two species were
sampled, especially in Qaidam Basin and the northern margin of TP. The top five families with the largest



number of sampled species were as follows: Asteraceae (83 species and 24 genera), Poaceae (47 species
and 18 genera), Fabaceae (46 species and 11 genera), Cyperaceae (29 species and four genera) and
Rosaceae (28 species and 10 genera). Amongst the 468 species, 79 species were unique to the TP, such
as *Rhodiola smithii*, *Pomatosace filicula*, *Oxytropis sericopetala*, *Arenaria gerzensis*, *Onosma waltonii*,
*Delphinium qinghaiense*, *Metaeritrichium microuloides* and *Androsace cuttingii*. Furthermore, two were
endangered species (*Rosa rugosa* and *Rheum globulosum*), seven were vulnerable species (*Arnebia*
*guttata*, *Tamarix taklamakanensis*, *Rhodiola smithii*, *Juniperus tibetica*, *Reaumuria kaschgarica*, *Rheum*
*tanguticum* and *Metaeritrichium microuloides*) and 10 were near-threatened species (*Myricaria prostrata*,
*Euphorbia kozlovii*, *Hippophae tibetana*, *Phlomis pygmaea*, *Physochlaina praealta*, *Gentiana*
*siphonantha*, *Astragalus handelii*, *Androsace cuttingii*, *Carex nakaoana* and *Leiospora exscapa*) in the
TiP-Leaf dataset.
**5.2 Leaf trait variations**
The site-level leaf traits are shown in Table 1. The variation of each leaf trait was significant. In
particular, DW, FW and LA varied by more than 150%, followed by LT and LWC. The variations of
LMA, SLA, LDMC, LCC, LNC and LPC were slightly stable.
**Table 1** Summary of leaf functional traits in the TiP-Leaf dataset.

| Traits | Mean ± SD | Max | Min | CV (%) |
|---|---|---|---|---|
| LT (mm) | 0.42 ± 0.22 | 1.55 | 0.06 | 52.38 |
| FW (g) | 0.14 ± 0.31 | 3.82 | 0.0001 | 221.43 |
| DW (g) | 0.04 ± 0.07 | 0.62 | 0.00003 | 175.00 |
| LDMC (g·g$^{-1}$) | 0.37 ± 0.09 | 0.75 | 0.08 | 24.32 |
| LWC (g·g$^{-1}$) | 2.41 ± 1.25 | 11.11 | 0.33 | 51.87 |
| LA (cm$^2$) | 3.22 ± 5.23 | 44.51 | 0.01 | 162.42 |
| SLA (cm$^2$·g$^{-1}$) | 142.10 ± 46.67 | 333.85 | 33.16 | 32.84 |
| LMA (g·m$^{-2}$) | 91.49 ±34.28 | 308.11 | 17.22 | 37.36 |
| LCC (mg·g$^{-1}$) | 386.54 ± 43.46 | 487.42 | 212.57 | 11.24 |
| LNC (mg·g$^{-1}$) | 23.08 ± 4.75 | 45.83 | 7.18 | 20.58 |
| LPC (mg·g$^{-1}$) | 1.62 ± 0.51 | 3.76 | 0.43 | 31.48 |





**5.3 Leaf trait relationships**
The fitting of linear models of the site averages of six leaf traits (Fig. 3) showed that LT was
significantly negatively correlated with LDMC ($r^2 = 0.1339$; $p < 0.05$; Fig. 3a), SLA ($r^2 = 0.0533$; $p<0.05$;
Fig. 3b) and LCC ($r^2 = 0.2755$; $p < 0.05$; Fig. 3c), with downward trends. No relationship was found
between LT and LNC ($r^2 = -0.0028$; $p > 0.05$; Fig. 3d) nor LPC ($r^2 = 0.0055$; $p > 0.05$; Fig. 3e). The
results also revealed that LDMC was significantly negatively correlated with SLA ($r^2 = 0.0630$; $p < 0.05$;
Fig. 3f), LNC ($r^2 = 0.0335$; $p < 0.05$; Fig. 3h) and LPC ($r^2 = 0.0649$; $p < 0.05$; Fig. 3i) and significantly
positively correlated with LCC ($r^2 = 0.0526$; $p < 0.05$; Fig. 3g). In addition, linearly inversed relationships
were observed between SLA and LCC ($r^2 = 0.0371$; $p < 0.05$; Fig. 3j), and LNC ($r^2 = 0.0566$; $p < 0.05$;
Fig. 3k) and LPC ($r^2 = 0.1776$; $p < 0.05$; Fig. 3l). The three leaf chemical traits were also related to one
another and the relationship between LNC and LPC ($r^2 = 0.2481$; $p < 0.05$; Fig. 3o) was closer than that
between LNC and LCC ($r^2 = 0.0552$; $p < 0.05$; Fig. 3m) and between LPC and LCC ($r^2 = 0.0879$; $p <$
0.05; Fig. 3n).

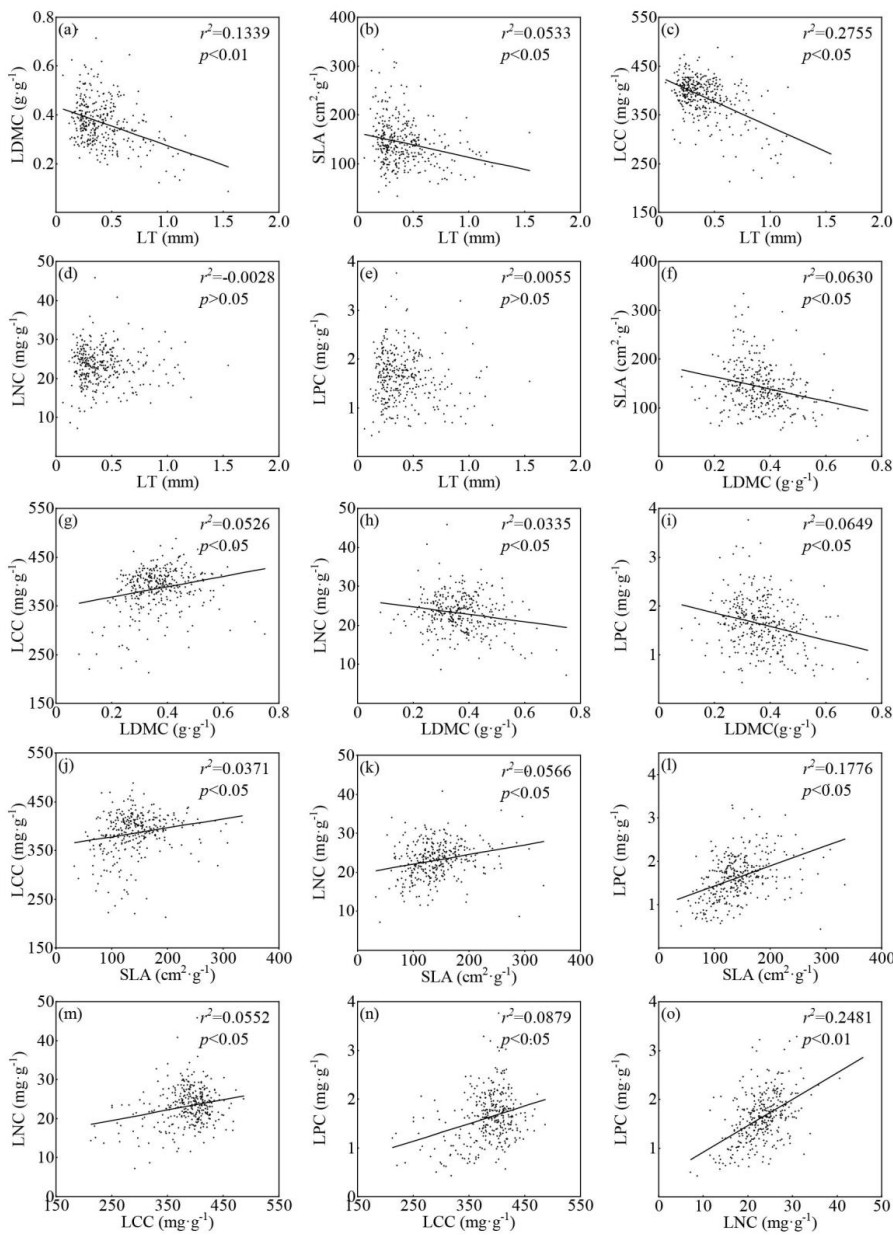

**Figure 3.** Relationship between site-based average of key leaf traits in the TiP-Leaf dataset. The black dot indicates the mean leaf trait measurements of all species in the site and the straight line represents the fitting of the linear model. $r^2$ is the adjusted $r^2$ and $p$ represents the probability value of the regression model.

**6 Data availability**

The TiP-Leaf dataset includes three data sheets in Microsoft Excel format, namely: (a) a data sheet




named 'variables' describing header information of the geographical coordinates, climate and traits in
the dataset (Table 2); (b) a data sheet (site information) reporting the site location and climate data; and
(c) a data sheet (plant traits) of the complete trait data of each plant species in each sampling site. As
studies based on the TiP-Leaf dataset are already underway, researchers interested in using such data
previously are strongly recommended to contact the authors to avoid overlapping studies. The dataset
will be available through the National Tibetan Plateau Data Center (TPDC; Jin et al., 2022;
https://doi.org/10.11888/Terre.tpdc.272516), and shall also be made available via the global TRY plant
trait database (Kattge et al., 2011, 2020; www.try-db.org/).

**Table 2** Summary information found in TiP-Leaf dataset

| Heading | Description | Type |
|---|---|---|
| Site | Site number based on sampling time | Code |
| Lat | Latitude (decimal degrees) | Numeric |
| Lon | Longitude (decimal degrees) | Numeric |
| Elev | Elevation (m) | Integer |
| Animal intensity | Animal activity intensity | Character |
| Human intensity | Human interference intensity | Character |
| Vegetation type | Vegetation type from field survey | Character |
| Vegetation region | Vegetation region from vegetation map | Character |
| MAT | Mean annual temperature (°C) | Numeric |
| MTCO | Mean temperature of the coldest month | Numeric |
| MTWA | Mean temperature of the warmest month | Numeric |
| $GDD_0$ | Annual growing degree days above 0 °C | Numeric |
| $GDD_5$ | Annual growing degree days above 5 °C | Numeric |
| MAP | Mean annual precipitation (mm) | Numeric |
| GP | Growing season precipitation (mm) | Numeric |
| MI | Moisture index | Numeric |
| DI | Drought index | Numeric |
| Species | Scientific name | Character |
| Family | Botanical family | Character |



| Growth form | Trees, shrubs, semi-shrubs and herbs | Character |
| --- | --- | --- |
| Life form | Deciduous and evergreen; Annual and perennial | Character |
| LT | Leaf thickness (mm) | Numeric |
| FW | Fresh weight (g) | Numeric |
| DW | Dry weight (g) | Numeric |
| LDMC | Leaf dry-matter content ($g \cdot g^{-1}$) | Numeric |
| LWC | Leaf water content ($g \cdot g^{-1}$) | Numeric |
| LA | Leaf area ($cm^2$) | Numeric |
| SLA | Specific leaf area ($cm^2 \cdot g^{-1}$) | Numeric |
| LMA | Leaf mass per area ($g \cdot m^{-2}$) | Numeric |
| LCC | Leaf carbon concentration ($mg \cdot g^{-1}$) | Numeric |
| LNC | Leaf nitrogen concentration ($mg \cdot g^{-1}$) | Numeric |
| LPC | Leaf phosphorus concentration ($mg \cdot g^{-1}$) | Numeric |

**7 Summary**

The TiP-Leaf dataset was compiled from direct field measurements, covering a great proportion of plant species and vegetation types on the highest plateau in the world. The dataset provides important data foundation not only for quantitative analyses of modern alpine vegetation but also for prediction of future response of alpine ecosystem to climate change and improvement of next-generation vegetation models. It could also be used to promote the vegetation protection and restoration on the TP and contribute to the global plant trait database.

The dataset in this study provides more leaf trait measurements and covers more sampling sites, which were located not only along the main roads but also the accessible small paths, than previous studies (Luo et al., 2005; He et al., 2006; He et al., 2010; Geng et al., 2014; Wang et al., 2020; Xu et al., 2021). This dataset is the first plant trait dataset that represents all of the alpine vegetation on the TP. However, more collections of trait data are needed in remote areas with assessable difficulty, such as the Hoh Xil dead zone in Northwest TP (alpine meadow, steppe and desert vegetation) and in the mountainous areas of East and Southeast TP with less trait studies (subalpine and alpine forest and shrubland vegetation). These works could enhance the representativeness of the whole TiP trait database





in terms of geographical space and vegetation type. Given the complex topography of the plateau, more
sites are requested to be surveyed and given the flourish of alpine flora, traits from more plant species
should be measured. The TiP-Leaf dataset consists of leaf traits only. The TiP-Root trait dataset is
underway and the trait data of twig and branch of woody species should be further measured.
**Author contributions.** JN conceived the study. KL, YJ and JN led the field works. YJ and other co-
authors collected leaf samples and measured plant traits. YJ, HW and JX processed the dataset, performed
the analyses and wrote the first draft. JN and YJ improved the manuscript. All authors approved the final
version of the submitted manuscript.
**Competing interests.** The (co-) authors declare that they have no conflict of interest.
**Disclaimer.** Publisher's note: Copernicus Publications remains neutral with regard to jurisdictional
claims in published maps and institutional affiliations.
**Acknowledgements.** This work was funded by the Second Tibetan Plateau Scientific Expedition and
Research Program (STEP, 2019QZKK0402) and the Strategic Priority Research Program of the Chinese
Academy of Sciences (XDA2009000003). The authors sincerely thank Chenyu Li, Tudan Luosang, Yezi
Sheng, Pingyu Sun and Deyu Xu for their help in the field survey and Jun Li, Ang Liu, Rui Tang and
Xinxin Zhou for helping with specimen identification.
**Financial support.** This work was supported by the Second Tibetan Plateau Scientific Expedition and
Research Program (STEP, 2019QZKK0402) and the Strategic Priority Research Program of the Chinese
Academy of Sciences (XDA2009000003).

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
