# Peer review of "TiP-Leaf: A dataset of leaf traits across vegetation types"

_Earth System Science Data, 2022_

## Author Comment (AC1)

Dear reviewer,

Thank you very much for reviewing our manuscript (essd-2022-199). We appreciate your approbation to the manuscript and also your valuable comments. We response to your comments one by one as follows:

**1. Line 141-142: Please elaborate on what you define as the "degree of drought" to further subclassify the vegetation regions. If you refer to the Annual Drought Index or any other parameter (mentioned below in the M&Ms), please explain this to make it clear to the reader.**

This is based on the official vegetation regionalization of China. The "degree of drought" usually referred to a drought index, the Selianinov drought index, in the Vegetation of China and the Vegetation Regionalization Map of China. The drought index K=0.16*accumulated temperature of a year (>10°C)/precipitation of the period when temperature >10°C. It is not the same factor as used in the manuscript used (the Drought Index appeared later).

To make it clear to the reader, we modified the text: Therefore, in accordance with the degree of drought, the Selianinov drought index used in the Vegetation Regionalization Map of China (ECVMC, 2007b), TP vegetation was further divided into three subregions from southeast to northwest: East TP Alpine Scrub and Alpine Meadow Subregion, Middle TP Alpine Steppe Subregion and Northwest TP Alpine Desert Subregion.

**2. Line 190-194, Data analysis: I am missing the description of the "statistical analyses" mentioned in this paragraph as well as the packages and software used to make the analyses. What was the purpose of making the linear models at the site level? Please elaborate.**

**Also, why did you use the averages of the leaf traits per site when you have a variety of life forms and life strategies, which will translate into contrasting differences in the leaf traits, particularly in the morphological ones?**

First of all, we would like to say that the manuscript is a data description paper, which focusses more on data description rather than data analyses. However, some simple statistical analyses have been conducted too. The site-based traits were simply analyzed and mapped using the Origin 2022, which was indeed not noted in the "Data analysis". Simple linear models were used to assess the relationships among key leaf traits at the site level to reveal the trade-off between different traits in the special alpine ecosystem.

Furthermore, the purpose of our data description manuscript is to provide readers with a general pattern of trait relationships. Thus, we choose to quantify the trait relationships at the averaged site level rather than in functional groups. We are using the trait data to do more analyses, and a manuscript entitled "The unique pattern and variation mechanism in key leaf traits on the Tibetan

Plateau" is being prepared and will be submitted soon. All of the features of leaf traits of the Tibetan ecosystems, their variations and relationships in growth forms, life forms at species and site levels, are all being analyzed.

Anyway, to make the purpose of the statistical analyses clearer, we improved the subsection "3.4 Data analysis": Beside the data description of leaf trait characteristics, six key leaf functional traits (LT, LDMC, SLA, LCC, LNC and LPC), which reflect the key ecological significances of plants grew in high altitude and extremely cold environment, were selected in this paper for further simple statistical analyses. The mean, minimum, maximum, standard deviation (SD) and coefficient variation of traits at each site were calculated, to generally show the pattern of leaf traits of the Tibetan ecosystems. The linear relationships between leaf traits of site average were analysed and mapped using the Origin software (The Origin Lab, 2022?). The detailed analyses of all of the leaf traits, their variations and spatial patterns, within and among functional groups and at species and site levels, will be further analysed in another paper.

**3. Section 5.3 Leaf trait relationships (L 248-260): Could you explain why you have selected to analyze the relationships between these traits? For instance, it would be nice to see some explanation (even if it is a brief one) linking this aspect of variation in certain traits (e.g., LT, LDMC, LMA) to the mechanisms of variation among different plant functional types and environments surveyed in this study.**

The six fundamental leaf traits we selected for analyze the relationships due to their ecological significance in the face of high altitude and extremely cold environment (have been added in subsection 3.4 mentioned above): LT, affecting the water supply and storage of leaves and the exchange process of matter and energy in photosynthesis; LDMC, reflecting the ability of plants to acquire surrounding environmental resources; SLA, considered as the first choice index for studying plant physiological and ecological strategies under specific environmental conditions; LCC, the main structural material of plants; LNC, characterizing the ability of plants to absorb and utilize nutrient elements; and LPC, the second largest element affecting plant growth. Again, the main aim of this paper is to present some basic information associated with leaf trait dataset to readers. The detailed analyses linking trait variation to environmental variables among different functional types are being analysed in another paper.

**4. Lastly, I greatly encourage the writers to revise the usage of the English language.**

The submitted manuscript has been check its usage of the English language by a native English speaker (whose background is likely biology). During the revision, we will seek for another expert from the research field of ecology to further improve the English language of the manuscript.

**5. As for the data set (Excel file), I suggest the following minor corrections: Line 55, for the species Artemisia frigida in Site TP2018080501. Could you explain why it is classified as a "Coniferous forest"? I would suggest that the term "alpine scrubland" is more appropriate according to the vegetation types and floristic composition described for this site. Also, the elevation seems quite high for a forest.**

**As a side note, is there any information on the soil types at each of the sampling sites? Because that would also be useful information to fully understand the floristic composition and distribution for people not familiar with Chinese flora.**

We have carefully checked the sampling site (2018080501), and there is no *Artemisia frigida*. *Juniperus convallium* is the dominant species in this site so the vegetation type was classified as "coniferous forest", a subalpine coniferous forest. The Vegetation of China described that on sunny slopes f the Tibetan Plateau below 4500 m in altitude with good environmental conditions, some big alpine and subalpine forests such as *Juniperus tibetica* or *Juniperus convallium* are developed Thus, it is convincing that the vegetation type of this site we investigated belongs to coniferous forest.

As for the information of soil types at each sampling point, we will extract it from the existing database of Soils of China. We actually measured the soil properties such the elemental contents of CNP, but these data will be using in another paper.

All of your helpful comments will be considered accordingly in revising the manuscript. Please do not to hesitate to contact us, if you have any further questions about the manuscript and the answers.

Sincerely yours,
Yilin Jin and Jian Ni
On behalf of all the co-authors

---

## Author Comment (AC2)

Dear reviewer,

We sincerely thank for your review on our manuscript. According to your suggestions, we have made some changes to our manuscript. The detailed corrections are listed below.

*1 I recommend the authors to describe the study area with "Qinghai-Tibetan Plateau".*

Thanks for the suggestion. We have noted that several words about the name of the plateau existed, for example, the Tibetan Plateau, Qinghai-Tibetan Plateau, Qingzang Plateau and even Xizang Plateau. The first word has often been used internationally, and the other three have usually been used by Chinese scientists. We looked through recent papers published from ESSD, the word "Tibetan Plateau" was more often used than the Qinghai-Tibetan Plateau. It is also called the "Tibetan Plateau" in the special issue "Extreme environment datasets for the three poles" which our manuscript submitted. In order to keep unanimous with the special issue, we still use the word "Tibetan Plateau" but explained in the Introduction that the TP is call "Qinghai-Tibetan Plateau" too and it includes two regions of Qinghai and Xizang in China.

*2 I agree that some datasets have built in China, but it has no data on the QTP. However, some progresses about trait analysis have been achieved and the description of these work is necessary.*

In the section ("Introduction") of our manuscript we have generally introduced some trait-based works carried out on the Tibetan Plateau but lacked detailed description. Thus, we have made some corresponding adjustments to this part. Details are as follows:

"Field-based, small regional studies of plant functional traits on the TP had made some interesting advances, such as Luo *et al.* (2005) linked the plant traits with ecosystem functions, He *et al.* (2006) explored the influence factors on plant traits, Geng *et al.* (2014) quantified the patterns of plant trait correlations between above- and below-ground components, Wang *et al.* (2020) compared their work with global dataset, and Xu *et al.* (2021) analyzed the mechanism of plant trait variation along the altitude pattern. Yet such works were also limited, where the sampling sites have been mostly along the main roads in East TP. Plant trait records from Central to West TP are very rare."

*3 Line 102, please added the description of sampling size in your dataset.*

We have supplemented the relevant description in line 104. "The dominant and common plant species in each site were determined by visual inspection, and leaf samples of these plants were picked up and the leaf traits were measured."

*4 Line 172, please add the method reference of measuring LNC and LPC.*

Determination of nitrogen and phosphorus in plants was conducted based on the agricultural industry standards of the People's Republic of China (NY/T 2017-2011, Fang *et al.*, 2011).

Fang JB, Pang RL, Guo LL, Xie HZ, Li J, Luo J, Yu H, Liu Y, Wu FK. 2011. Determination of

nitrogen, phosphorus and potassium in plants. NY/T 2017–2011.

*5 The ecological significance of these traits should be added.*

We have added this information in our manuscript (3.4 Data analysis). Details are as follows:

"The six fundamental leaf traits we selected for analyze the relationships due to their ecological significance in the face of high altitude and extremely cold environment: LT, affecting the water supply and storage of leaves and the exchange process of matter and energy in photosynthesis; LDMC, reflecting the ability of plants to acquire surrounding environmental resources; SLA, considered as the first choice index for studying plant physiological and ecological strategies under specific environmental conditions; LCC, the main structural material of plants; LNC, characterizing the ability of plants to absorb and utilize nutrient elements; and LPC, the second largest element affecting plant growth."

*6 The potential applications and the limitations (if some uncertainties existed during measuring) should be discussed.*

The previous version of our manuscript has briefly discussed the potential applications and limitations of the dataset in the "Summary" section, but some uncertainties existed during measuring have been absent. In the current version, we added more detailed descriptions of this part in the first paragraph of the "Summary" section. Details are as follows:

"The TiP-Leaf dataset was compiled from direct field measurements, covering a great proportion of plant species and vegetation types on the highest plateau in the world. The dataset provides important data foundation not only for quantitative analyses of modern alpine vegetation but also for prediction of future response of alpine ecosystem to climate change and improvement of next-generation vegetation models. It could also be used to promote the vegetation protection and restoration on the TP and contribute to the global plant trait database. However, the dataset also presents some unavoidable limitations. For example, the establishment of sampling sites and the judgment of dominant and common species are mostly subjective. The leaves of some plants are extremely small, resulting in incomplete recognition when scanning the leaf area. Due to the harsh field conditions, it is sometimes impossible to determine the plant traits in time. It is still impossible to prevent some leaves from losing too much water to withering although we have taken protective measures for the leaves. Inadequate collection of some leaf samples results in less data of plant chemical element content than that of morphological traits. In any case, it is not easy to carry out large-scale collection of plant traits on the Tibetan Plateau, which requires a lot of manpower and material resources as well as overcoming the adverse environment of high altitude and extreme variability."

Please do not to hesitate to contact us, if you have any further questions about the manuscript and the answers.

Sincerely yours,

Yilin Jin and Jian Ni

On behalf of all the co-authors

---

## Author Response (AR1)

**Response to the Referees**

(TiP-Leaf: A dataset of leaf traits across vegetation types on the Tibetan Plateau)

(essd-2022-199)

**Dear Dr. Carlson,**

Thank you and the reviewers again for providing very helpful comments on our manuscript. We have carefully considered every comment and make accordingly detailed revision. All revised contents have been highlighted by red colors in the marked-up version of revised manuscript. Furthermore, the point-to-point responses have also been presented here. The replies have been already posted in the interactive comments, here we put all of the replies together and will posted them again.

We hope this revision will meet the requirement for publication in *Earth System Science Data*.

Sincerely Yours,

Yili Jin (1844044255@qq.com) and Jian Ni (nijian@zjnu.edu.cn)

On behalf of all the co-authors
* * *
**# Referee report 1**

*The manuscript titled "TiP-Leaf: A dataset of leaf traits across vegetation types on the Tibetan Plateau" provides relevant information to further understand the floristic composition and leaf trait variation for the vegetation in the Tibetan Plateau. In itself, the monumental task of measuring leaf traits and compiling the data for the species from 336 sites across the Tibetan Plateau is to be appreciated and will greatly benefit the scientific community by filling out information gaps at the regional scale in further studies or global metanalysis. However, I believe there are some points in the structure and content of the current manuscript that need to be improved. To this end, please find my specific comments below.*

**Response:** We sincerely appreciate your approbation to the manuscript and your valuable comments. Please refer to replies as following.

**Comment 1:** *Line 141-142: Please elaborate on what you define as the "degree of drought" to further subclassify the vegetation regions. If you refer to the Annual Drought Index or any other*

*parameter (mentioned below in the M&Ms), please explain this to make it clear to the reader.*

**Response:** This is based on the official vegetation regionalization of China. The "degree of drought" usually referred to a drought index, the Selianinov drought index, in the Vegetation of China and the Vegetation Regionalisation Map of China. The drought index K=0.16*accumulated temperature of a year (>10 °C)/precipitation of the period when temperature >10 °C. It is not the same factor as used in the manuscript used (the Drought Index appeared later). To make it clear to the reader, we modified the text as following.

**On Line 154-158 (**in the marked-up version of revised manuscript)**.** "Therefore, in accordance with the degree of drought, the Selianinov drought index used in the Vegetation Regionalisation Map of China (ECVMC, 2007b), TP vegetation was further divided into three subregions from southeast to northwest, namely, East TP Alpine Scrub and Alpine Meadow Subregion, Middle TP Alpine Steppe Subregion and Northwest TP Alpine Desert Subregion."

*Comment 2: Line 190-194, Data analysis: I am missing the description of the "statistical analyses" mentioned in this paragraph as well as the packages and software used to make the analyses. What was the purpose of making the linear models at the site level? Please elaborate. Also, why did you use the averages of the leaf traits per site when you have a variety of life forms and life strategies, which will translate into contrasting differences in the leaf traits, particularly in the morphological ones?*

**Response:** First of all, we would like to say that the manuscript is a data description paper, which focusses more on data description rather than data analyses. However, some simple statistical analyses have been conducted too. The site-based traits were simply analyzed and mapped using the Origin 2022, which was indeed not noted in the "Data analysis". Simple linear models were used to assess the relationships among key leaf traits at the site level to reveal the trade-off between different traits in the special alpine ecosystem. Furthermore, the purpose of our data description manuscript is to provide readers with a general pattern of trait relationships. Thus, we choose to quantify the trait relationships at the averaged site level rather than in functional groups. We are using the trait data to do more analyses, and a manuscript entitled "The unique pattern and variation mechanism in key leaf traits on the Tibetan Plateau" is being prepared and will be submitted soon. All the features of leaf traits of the Tibetan ecosystems, their variations and relationships in growth forms, life forms at species and site levels, are all being analyzed. Anyway, to make the purpose of the statistical analyses clearer, we improved the subsection "3.4 Data analysis".

**On Line 213-218.** "The mean, minimum, maximum, standard deviation (SD) and coefficient variation of traits at each site were calculated to generally show the pattern of leaf traits of the Tibetan ecosystems. The linear relationships between leaf traits of site average were analysed and

mapped using the Origin software (The Origin Lab, 2022) to reveal the trade-off between different traits in the special alpine ecosystem. The detailed analyses of all the leaf traits, their variations and spatial patterns, within and amongst functional groups and at species and site levels, will be further analysed in another paper."

*Comment 3: Section 5.3 Leaf trait relationships (Line 248-260): Could you explain why you have selected to analyze the relationships between these traits? For instance, it would be nice to see some explanation (even if it is a brief one) linking this aspect of variation in certain traits (e.g., LT, LDMC, LMA) to the mechanisms of variation among different plant functional types and environments surveyed in this study.*

**Response:** The six fundamental leaf traits we selected for analyze the relationships due to their ecological significance in high altitude and extremely cold environment. Again, the main aim of this paper is to present some basic information associated with leaf trait dataset to readers. The detailed analyses linking trait variation to environmental variables among different functional types are being analysed in another paper.

**On Line 205-213.** "Beside the data description of leaf trait characteristics, six key leaf functional traits (e.g. LT, LDMC, SLA, LCC, LNC and LPC), that reflect the key ecological significances of plants that grew in high altitude and extremely cold environment, were selected in this paper for further simple statistical analyses. LT affects the water supply and storage of leaves and the exchange process of matter and energy in photosynthesis; LDMC reflects the ability of plants to acquire surrounding environmental resources; SLA is considered the first choice index for studying plant physiological and ecological strategies under specific environmental conditions; LCC is the main structural material of plants; LNC characterises the ability of plants to absorb and utilise nutrient elements; and LPC is the second largest element affects plant growth."

*Comment 4: Lastly, I greatly encourage the writers to revise the usage of the English language.*

**Response:** The previous submitted manuscript has been checked its usage of the English language by a native English speaker (whose background is likely biology). Now, we have sought for another expert from the research field of ecology to further improve the English language of the manuscript. All traces of new revisions are highlighted in the red color, please check the marked-up version of revised manuscript.

*Comment 5: As for the data set (Excel file), I suggest the following minor corrections: Line 55, for the species Artemisia frigida in Site TP2018080501. Could you explain why it is classified as a "Coniferous forest"? I would suggest that the term "alpine scrubland" is more appropriate*

*according to the vegetation types and floristic composition described for this site. Also, the elevation seems quite high for a forest. As a side note, is there any information on the soil types at each of the sampling sites? Because that would also be useful information to fully understand the floristic composition and distribution for people not familiar with Chinese flora.*

**Response:** We have carefully checked the sampling site (2018080501), and there is no *Artemisia frigida*. *Juniperus convallium* is the dominant species in this site so the vegetation type was classified as "coniferous forest", a subalpine coniferous forest. The Vegetation of China described that on sunny slopes of the Tibetan Plateau below 4500 m in altitude with good environmental conditions, some big alpine and subalpine forests such as *Juniperus tibetica* or *Juniperus convallium* are developed. Thus, it is convincing that the vegetation type of this site we investigated belongs to coniferous forest. As for the information of soil types at each sampling point, we will extract it from the existing database of Soils of China and upload it as a supplement. We actually measured the soil properties such the elemental contents of CNP, but these data will be using in another paper.

**Referee report 2**

*The authors have submitted an outstanding dataset of plant traits on the data limited Qinghai-Tibetan Plateau. I am certain that this data will be very valuable and reliable to the scientific community of plant ecologists and beyond in this region. While the value of these data is unquestionable, I have some questions with the data description.*

**Response:** We sincerely thank for your review on our manuscript. According to your comments, we have made some changes to our manuscript. The detailed corrections are listed below.

***Comment 1:** I recommend the authors to describe the study area with "Qinghai-Tibetan Plateau".*

**Response:** Thanks for the suggestion. We have noted that several words about the name of the plateau existed, for example, the Tibetan Plateau, Qinghai-Tibetan Plateau, Qingzang Plateau and even Xizang Plateau. The first word has often been used internationally, and the other three have usually been used by Chinese scientists. We looked through recent papers published from ESSD, the word "Tibetan Plateau" was more often used than the Qinghai-Tibetan Plateau. It is also called the "Tibetan Plateau" in the special issue "Extreme environment datasets for the three poles" which our manuscript submitted. In order to keep unanimous with the special issue, we still use the word "Tibetan Plateau" but explained in the Introduction that the TP is call "Qinghai-Tibetan Plateau" too and it includes the two major regions of Qinghai and Xizang in China.

**On Line 50-54.** "The Tibetan Plateau (TP), includes the two major regions of Qinghai Province and Xizang Autonomous Region, and partial areas from northwestern Gansu Province, western Sichuan Province and northwestern Yunnan Province in China, is also called 'Qinghai-Tibetan Plateau'. As

the world's 'Third Pole', 'Asia Water Tower' and the cradle of the East Asian flora, TP is the most under-representative region in global and regional plant trait databases."

**Comment 2:** *I agree that some datasets have built in China, but it has no data on the QTP. However, some progresses about trait analysis have been achieved and the description of these work is necessary.*

**Response:** In the section ("Introduction") of our manuscript we have generally introduced some trait-based works carried out on the TP but lacked detailed description. Thus, we have made some corresponding adjustments to this part.

**On Line 57-65.** "Field-based, local studies of plant functional traits on the TP had made some interesting advances. For example, Luo et al. (2005) linked the plant traits with ecosystem functions, He et al. (2006) explored the influencing factors on plant traits, Geng et al. (2014) quantified the patterns of plant trait correlations between above- and below-ground components, Wang et al. (2020) compared their work with global dataset, and Xu et al. (2021) analysed the mechanism of plant trait variation along the altitude pattern. Yet such works, where the sampling sites have been mostly along the main roads in East TP, were also limited."

**Comment 3:** *Line 102, please added the description of sampling size in your dataset.*

**Response:** We have supplemented the relevant description.

**On Line 115-116.** "The dominant and common plant species in each site were determined by visual inspection, the leaf samples of these plants were picked up, and the leaf traits were measured."

**Comment 4:** *Line 172 please add the method reference of measuring LNC and LPC.*

**Response:** Determination of nitrogen and phosphorus in plants was conducted based on the agricultural industry standards of the People's Republic of China (NY/T 2017-2011, Fang et al., 2011). In addition, we have also added the reference of LCC determination.

**On Line 185-187.** "LCC (mg·g$^{-1}$), LNC (mg·g$^{-1}$) and LPC (mg·g$^{-1}$) were determined by outside-temperature hot potassium dichromate oxidation–volumetric method (Wu, 2007), distillation–titration method and vanadium molybdate yellow colorimetric method (Fang et al., 2011), respectively."

**On Line 461-462.** "Fang, J. B., Pang, R. L., Guo, L. L., Xie, H. Z., Li, J., Luo, J., Yu, H., Liu, Y., and Wu, F. K.: Determination of nitrogen, phosphorus and potassium in plants (NY/T 2017–2011), 2011."

**On Line 739-740.** "Wu, D. M.: Protocols for Standard Biological Observation and Measurement in Terrestrial Ecosystems, China Environmental Science Press, Beijing, 2007."

***Comment 5:*** *the ecological significance of these traits should be added.*

**Response:** We have added this information in our manuscript (3.4 Data analysis), as suggested by another manuscript reviewer.

***Comment 6:*** *the potential applications and the limitations (if some uncertainties existed during measuring) should be discussed.*

**Response:** The previous version of our manuscript has briefly discussed the potential applications and limitations of the dataset in the "Summary" section, but some uncertainties existed during measuring have been absent. In the current version, we added more detailed descriptions of this part in the first paragraph of the "Summary" section.

**On Line 312-321.** "However, the dataset also presents some unavoidable limitations. For example, the establishment of sampling sites and the judgment of dominant and common species are mostly subjective. The leaves of some plants are extremely small, resulting in incomplete recognition when scanning the LA. Due to harsh field conditions, measuring the plant traits in time occasionally becomes impossible. Prevent some leaves from losing too much water to withering is still inevitable, although we have taken protective measures for the leaves. Inadequate collection of some leaf samples results in less data of plant chemical element content than that of morphological traits. In any case, performing large-scale collection of plant traits on the TP, which requires a lot of manpower and material resources, as well as overcoming the adverse environment of high altitude and extreme variability, is not easy."

---

## Referee Report (RR1)

All my questions were thoroughly resolved and elaborated, and I recommend publication of the manuscript.

---

## Author Response (AR2)

**Response to the Comments**

(TiP-Leaf: A dataset of leaf traits across vegetation types on the Tibetan Plateau)

(essd-2022-199)

**Dear Dr. Carlson,**

Thank you very much for approving our manuscript and deciding to publish it in *Earth System Science Data*! All revised contents have been highlighted by the red color in the marked-up version of revised manuscript. Additionally, the point-to-point responses have also been presented here.

Please do not hesitate to contact us if you have further questions on our final manuscript.

Sincerely Yours,

Yili Jin (1844044255@qq.com) and Jian Ni (nijian@zjnu.edu.cn)
On behalf of all the co-authors
* * *
**# Comments to the author:**

*Interesting data set from a challenging region. Thank you for using ESSD!*

**Response:** Thank you again sincerely for accepting our manuscript for publication in *ESSD*!

**Comment 1:** *Numerous residual examples of mis-translation, Chinese to English. All small; language experts at Copernicus should catch most of these. Authors will need to check proofs very carefully.*

**Response:** Thanks to the language experts for checking the English language of our manuscript, and we will carefully check the proofs later when it is available.

**Comment 2:** *Not sure what authors mean at line 265: "slightly stable". Please rewrite in valid statistical terms.*

**Response:** What we originally wanted to explain was that compared with other traits (including DW, FW, LA, LT, and LWC) with extremely larger variation, the variations of these trait (LMA,

SLA, LDMC, LCC, LNC, and LPC) are relatively small. Perhaps our statement was not very clear, so we revised it again as following.

**On Line 263-265 (**in the marked-up version of revised manuscript). The site-level leaf traits are shown in Table 1. The variation of each leaf trait was significant. In particular, DW, FW and LA varied by more than 150%, followed by LT and LWC. The variations of LMA, SLA, LDMC, LCC, LNC and LPC were slightly  modest, relatively.

*Comment 3: With technical terms, abbreviations, and many species names, authors will - as mentioned above - check proofs very carefully.*

**Response:** We will check proofs very carefully about technical terms, abbreviations, and species Latin name, later when the proofs are available.